# Passive Soil Arching Effect in Aeolian Sand Backfills for Grillage Foundation

**DOI:** 10.3390/s23198098

**Published:** 2023-09-27

**Authors:** Chengcheng Zhang, Guanshi Liu, Shengkui Tian, Mingxuan Cai

**Affiliations:** 1State Key Laboratory of Geomechanics and Geotechnical Engineering, Institute of Rock and Soil Mechanics, Chinese Academy of Sciences, Wuhan 430071, China; chengcheng@163.com (C.Z.); kshengt@glut.edu.cn (S.T.); 2120210690@glut.edu.cn (M.C.); 2Guangxi Key Laboratory of Rock and Soil Mechanics and Engineering, Guilin University of Technology, Guilin 541004, China

**Keywords:** passive soil arching effect, trapdoor tests, influencing factors, grillage foundation, aeolian sand

## Abstract

The passive soil arching effect exists in many soil–grille interaction systems. Increasing mental grillage foundations are used for transmission lines in aeolian sand areas; thus, exploring the evolution mechanism of passive soil arching is crucial. This study investigates the evolution and influencing factors of passive soil arching through a series of tests using a trapdoor device and particle image velocimetry (PIV). The test results show that the evolution of the arching structure causes the aeolian sand deformation to gradually extend to the backfill surface and stationary zone, generating two triangular arching surfaces between the movable beams and sliding surface at the junction of the active and stationary zones. Cracks in the arching and sliding surfaces were connected to form a W-shaped shear band. The development of the soil pressure was divided into four arching structure stages. The different stages of the inner and outer arches of the bearing characteristics had strong differences. Taking the appearance of the first arch surface as the time point, the soil pressure changes abruptly and the inner and outer arches alternate to bear the as a major role. The beam spacing significantly affected the arching evolution. A smaller beam spacing formed an initial bending configuration with an inconspicuous arching structure and incomplete shear band. As the beam spacing increased, the arching shape changed from triangular to parabolic, sudden changes in the soil pressure were more pronounced, and the arch height increased. The relative density and water content had little impact on the arch shape and shear zone but significantly affected the arching strength, soil pressure transfer, and arching height. The medium and high relative densities and low water contents resulted in a stronger arching structure and greater arching height, while low relative densities and high water contents weakened the soil pressure transfer. The range values for the optimum beam spacing, relative density, and water contents are given based on the variation characteristics of the evaluated parameters (*E*, *n*) under different conditions.

## 1. Introduction

The aeolian sand area of northwest China is rich in natural mineral resources, such as oil and natural gas, and renewable resources, such as solar energy and wind energies. Thus, it is necessary to increase the number of transmission lines to meet resource development demands [1,2,3]. Grillage foundation is widely used for its convenient, fast construction and adaptability to harsh environments. The base plate is composed of a grillage made of steel or concrete beams [4,5] to produce obvious passive soil arching during uplift, significantly affecting the uplift of the foundation [6,7,8].

The soil arching effect is ubiquitous in soil–structure interaction systems [9]. The concept of arching effect was first introduced more than 100 years ago. Janssen [10] used a continuous medium model to quantitatively analyze the relationship between the force on a grain bin’s bottom surface and the grain accumulation size, demonstrating the arching effect. Robert [11] found that the load borne by the bottom of a barn remained constant after grain accumulation reached a certain height. Subsequently, Terzaghi [12] verified that the arching effect exists in soils employing a trapdoor test while proposing the primary conditions for its existence: (1) uneven settlement or relative displacement in soil, and (2) the arching foot can support the formation of soil arching. Schlosser [13] explained that the relative displacement between the soil and adjacent structures causes the soil arching effect. This interaction induces soil stress redistribution and load transfer to the underground structural elements, strengthening the bearing capacity and deformation resistance of the soil–structure system.

Since the soil arching effect was discovered, many scholars considered the mechanism of soil arching. Active and passive arching effects can be determined based on the relative displacement between the soil and structure [14,15]. Active arching is related primarily to the pile-supported subgrade [16,17] and shield tunnel [18,19]. Chen et al. [20] experimentally identified different arching evolutionary patterns and explored the relationship between the relative displacement and arching shape. Chevalier et al. [21] observed the evolution of a triangular expansion zone toward the vertical sliding zone through a single trapdoor test. Iglesia et al. [22] conducted embankment centrifugal tests and found that the evolution of soil arching can be divided into three stages, evolving from an initial curved configuration to a triangular shape, and eventually to a prismatic sliding mass. Rui et al. [23] proposed that the shape and development processes of soil arching follow three development patterns through two-dimensional (2D) trapdoor tests: tower development, triangle expansion, and similar settlement patterns. Hong et al. [24] verified that the soil arching effect was independent of the friction angle of backfill.

With continuous research on the soil arching effect, some influencing factors, such as the loading type [25,26] and backfill properties [27,28,29], have been considered. Sadrekarimi and Abbasnejad [30] determined the critical relative density (28%) for soil arching using concentric circular trapdoor tests with different diameters. Ma et al. [31] simulated the evolution of soil arching at different fill heights and used the load transfer efficacy to evaluate the structural development. They found that the load transfer efficacy and displacement of soil arching at stabilization increased with the backfill height. Bi et al. [32] investigated the evolution of the arching effect caused by cyclic loading, identified the evolution process of the arching from the initial formation to the final collapse, obtained stable and collapsed arching, and determined the critical loading amplitude corresponding to the threshold for the arching effect collapse. Zhang et al. [33] investigated the effects of static supercyclic surface loads and load frequencies on the soil arching effect. They found that the degradation of soil arching was more significant under cyclic loading than static loading. The interactions of buried structures, thin overburdened soils, and high-load frequencies accelerated soil arching degradation and led to greater surface settlement.

Some structures produce passive arching effects in many engineering applications, such as grillage foundations [34,35], cluster piles [36,37], and buried steel culverts [38]. Xiang REN et al. and Ma D et al. [39,40] studied the formation process and influence factors of passive soil arching before cantilever anti-slide piles through numerical modeling. However, this arch is horizontal rather than vertical. Khatami et al. [41] investigated the deformation and stress profiles of passive arching in coarse sand and two rubberized sand backfills using a trapdoor apparatus. Many scholars studied the influencing factors of passive soil arching, focusing on the trapdoor width and loading method, while fewer studies considered the relative density and water content. Zurita et al. [42] designed the beam spacing to ensure that the grillage foundation could form complete and continuous support to develop a passive soil arching structure while improving the foundation bearing capacity and deformation resistance. Xu et al. [43] conducted trapdoor tests to investigate soil arching behaviors under localized loading. They found that applying surface loading causes degraded soil arching, which was affected more by cyclic loading. Zhang et al. [44] analyzed passive arching under eccentric loading by developing a series of trapdoor discrete numerical models. Wu et al. and Zhao et al. [14,45] conducted alternating active and passive trapdoor tests to explore the mechanical behavior, kinematic mechanism, and progressive failure of the backfill subjected to continuous alternate motion.

Fewer studies exist on passive soil arching in the vertical direction, which needs further investigation. Therefore, a series of trapdoor model tests is performed by employing an electro-hydraulic servo-controlled loading system equipped with particle image velocimetry (PIV). The results help visualize and analyze the deformation of backfill sand and monitor soil pressure change to determine the evolution of passive soil arching and its influencing factors. These tests allow for a comprehensive understanding of the evolution and pressure development of passive soil arching. Finally, this study examines the impact of beam spacing, relative density, and water content on passive soil arching.

## 2. Materials and Methods

### 2.1. Testing Equipment

The test equipment includes a trapdoor apparatus and a comprehensive measurement system. The trapdoor apparatus comprises a sand chamber, trapdoor, and servo motor. The inner dimensions of the container are 500 mm × 250 mm × 600 mm (length × width × height), the side wall is made of transparent tempered glass, and the maximum filling height is 500 mm, as shown in Figure 1. The trapdoor comprises movable beams supported by steel beams and is set at the bottom of the sand chamber. The movable steel beams are 50 mm × 250 mm (length × width), and the width between adjacent movable beams can be freely adjusted. Moreover, the trapdoor apparatus is large enough relative to the stress test cell to ignore the effects of the size effect. A fibred seal covers both ends of the movable beams to prevent sand particles from getting between the edge of the movable beam and the fixed side wall. The servo motor is connected to the steel beam that supports the movable beams and is controlled by the motor system. The movable beam can be driven up at a speed of 0.03 mm/s with a maximum thrust of 10 kN and a maximum stroke of 60 mm. A stainless steel frame reinforcement supports the entire device.

The comprehensive measurement system includes a data acquisition system and a PIV system. The servo motor was equipped with a displacement transducer to record the trapdoor displacement with a maximum range of 60 mm and an accuracy of 0.01 mm. The miniature dynamic soil stress sensors (SSTs) with a diameter of 7.6 mm, thickness of 2 mm, resolution of 0.1%, sensitivity factor of 1%, and accuracy of 0.01 kPa were calibrated to a force range of 0–200 kPa (Model: PDA-PB) by the manufacturer (TML in Japan). These devices were mounted to the base of the trapdoor to measure stress changes. The PIV system (TSI) was mounted at the front of the sand chamber, capturing the deformation pattern of aeolian sand. A high-speed CCD camera with a resolution of 2456 × 2056 pixels and an accuracy of up to 0.02 pixels visualized the sand movement [46]. During the tests, the image processing software Insight 4GTM determined the displacement increments of the soil particles from continuous images. The software TECPLOT further processes data from Insight 4GTM. The details and principles of PIV are similar to that provided by Khatami and Wang [41,47].

### 2.2. Backfill Material

The backfills of the experiment were taken from the Yulin area (Northwest China, 38°19′02″ N, 109°39′09″ E) and belonged to the aeolian sand, which is unique to the southern margin of the Mu Us Desert. Figure 2 illustrates the particle size distribution. According to the Unified Soil Classification System (USCS), aeolian sand is poorly graded fine sand (SP) with a uniformity coefficient of Cu = 2.30 and curvature coefficient of Cc = 0.80. Table 1 summarizes the values of some other physical properties of the material, including direct shear tests for the shear parameters. The water content of aeolian sand is 2.8% under natural conditions. Table 2 and Table 3 show the mineral and chemical compositions of aeolian sand obtained via spectral testing (XRF) and X-ray diffraction (XRD), respectively.

### 2.3. Test Conditions

A total of 15 tests were conducted based on a division into four groups. Each group included four types of working conditions, as shown in Table 4. The design threshold (I) of the backfill height in the model was determined, and the geometric shape, displacement field, and soil pressure of passive soil arching were studied during the evolution process. The effects of the trapdoor width (II), relative density (III), and water content (IV) on the geometry and soil pressure evolution of passive soil arching were investigated. The group testing was divided as follows. Group I obtained the critical backfill height without deformation of the backfill surface. Group II determined the geometry of soil arching corresponding to different trapdoor widths. Groups III and IV provided the best relative density and water content for the formation of a stable soil arching effect.

### 2.4. Determination of Backfill Height

The model needs to consider the aeolian sand backfill height to ensure complete soil arching development and adequate load transfer. Chen et al. [20] simulated the working principle of an embankment using a trapdoor model test. They found that uneven settlements occur on the embankment surface when *H*/*s* ≤ 1.4 (*H* is the backfill height, and *s* is the trapdoor width). It is difficult to form complete soil arching, and when *H*/*s* ≥ 1.6, the surface of the embankment nearly does not produce a settlement, forming complete soil arching. There is a strong impact from fill height on the soil arching effect. This section explores the minimum backfill height threshold to form complete and stable soil arching with aeolian sand as the backfill material.

Figure 3 shows soil arching contours where the soil arching structure is formed at different backfill heights. However, at *H* = 200 mm (*H*/*s* ≥ 1.36), the displacement of the active zone increased and extended straight to the filling surface with larger uneven settlements and large cracks. Smaller cracks still appeared on the backfill surface at *H* = 300 mm (*H*/*s* = 2.4). At *H* ≥ 400 mm (*H*/*s* = 3.2), the overall displacement in the active zone was smaller, the impact was limited, and no cracks or significant settlements occurred at the backfill surface. It can be seen that when *H*/*s* ≥ 3.2, the backfill surface remains almost horizontal, and an equal settlement plane exists and beam subsoil relative displacement is restrained below the equal settlement plane [20,48].

Figure 4 shows variations in the average soil pressures on movable beams at different backfill heights. The average soil pressure on the movable beams changes significantly with displacement, showing that the pressures rapidly reached a maximum, the relative displacement only needed 2~3 mm, and the pressure tended to stabilize or decrease with the relative displacement. Comparing the maximum pressures on the movable beams shows that the soil pressures did not increase with the backfill height but stopped increasing after reaching a particular limiting value, similar to the “grain bin effect” [49].

Two evaluation parameters (load transfer efficacy [31,50] and stress concentration ratio [20,51]) are introduced to evaluate the evolution and load transfer of soil arching.
$$E=\frac{\sigma_{p}}{\sigma_{p}+\sigma_{s}}$$
$$n=\frac{\sigma_{s}}{\sigma_{p}}$$
where *E* is the load transfer efficacy, *n* is the stress concentration ratio, *σ_p_* is the average soil pressure on the movable beams, and *σ_s_* is the average soil pressure on the fixed beams. The more adequately the load transfers, the better the soil arching evolves, and the larger *E* and smaller *n* become.

Figure 5 illustrates the characteristic curves of the evaluation parameters for *E* and *n* at different backfill heights. Changes in the two parameters are similar to the soil pressure, where *E* and *n* reach their maximum values around *δ* = 4 mm and tend to stabilize thereafter. The *n* was relatively large at *H* = 200 mm (*H*/*s* = 1.6), but *E* was small. When *H* ≥ 300 mm (*H*/*s* ≥ 2.4), *n* decreased and stabilized at 0.12, while *E* increased and stabilized at 0.82, indicating that 82% of the overlying load was transferred to the movable beams. Moreover, when the fill height increased from 200 to 300 mm, *E* and *n* increased with a sharp slope with change rates of 16% and 15%, respectively. The *E* and *n* changed little as the fill height increased from 300 to 500 mm, with change rates of 1.3% and 0.8%, respectively (Figure 6). Thus, when *H*/*s* ≥ 2.4, the soil arching structure had a better load transfer efficacy and stress concentration ratio for the aeolian sand foundation. It can be seen that the backfill height has a significant influence on the evaluation parameters.

Variations in the load transfer efficacy and stress concentration ratio with the relative displacement and the development law of the soil arching structure at different backfill heights indicate that the backfill height for modeling tests can be set at 400 mm. Similarly, Khatami et al. [41] investigated passive arching in rubberized sand backfill using trapdoor tests and proposed a backfill height of not less than 250 mm.

### 2.5. Model Test Procedure

The model test procedures are as follows. (1) The movable beams are adjusted to obtain the required trapdoor width. (2) The six miniature dynamic SSTs are installed horizontally on one side of the sand chamber based on symmetry to measure the soil pressure distribution in the substrate. The SST locations are shown in Figure 1a, where SST1 and 2, are placed on the fixed beam between adjacent movable beams, SST3 and 4 are placed on the movable beams, and SST5 and 6 are on the fixed beams outside of the movable beams. All SSTs were connected to a Datataker-packed data acquisition device for data collection every 10 s. (3) The aeolian sand with different water contents is compacted to selected relative densities in the layers. The thicknesses of the first and second backfill sand layers on the trapdoor were 20 and 30 mm, and the rest were 50 mm. (4) The servo motor is controlled to drive the movable beams upward at a uniform speed of 0.03 mm/s, and the displacement is given on the controller screen. Friction on the sand chamber surface is about 0.06 kPa [52] and has little impact on the test results. (5) The evolution of soil arching is recorded using PIV. A CCD camera is fixed 0.6 m from the sand chamber according to the requirements of the close-up lens [53]. The height and focal length of the lens are adjusted to obtain a recording window of 450 mm × 250 mm (length × height), with the lowest position of the window aligned with the movable beams. The sampling rate of 10 frames per second is selected for automatic photography. The parameters of the test equipment are recorded and remain unchanged.

## 3. Results and Analysis

### 3.1. Evolution of Soil Arching Effect

Figure 7 shows the soil arching contours at different displacements obtained by PIV. The distortion of the fill material occurs primarily in the limited area above the trapdoor during soil arching development. As shown in Figure 7a, the movable beams translate upward, and the local aeolian sand particles squeeze before deforming, producing active and stationary zones in the backfill sand. When *δ* = 1.0 mm, the active zone of equal displacement spreads outward in an inverted U-shape, but deformation toward the horizontal sides gradually decreases. The average displacement is minimal at about 0.23 mm, the stress redistribution is not yet apparent, and only the soil arching prototype is produced between the movable beams. When *δ* = 4.0 mm, as shown in Figure 7b, the active zone expands in a wave shape to the stationary zone, and the sand particle displacement increases abruptly, resulting in the full development of stress redistribution and the formation of a triangular soil arching structure between the two active beams. Then, the arching structure begins to support the deformation and stress of the upper soil. The entire active zone is approximately rectangular.

When *δ* = 8 mm, as shown in Figure 7c, the movable beams continue to translate upward, and a substantial displacement difference occurs between the stationary and active zones. A tiny crack is generated at the critical interface between the active and stationary zones called the sliding surface and limits the range of the active zone [54]. The displacement of the entire active zone is reasonably uniform. Another crack appears between neighboring movable beams called the first arching surface (Figure 8a). The arching structure suffers minor damage but remains relatively stable. At this stage, the arching shape is more apparent and approximates an isosceles triangle, while the arching height is still rather small at approximately 67 mm. The active zone remains rectangular.

When *δ* = 12.0 mm, as shown in Figure 7d, the displacement in the active zone is significantly greater and nearly identical to that of the active beams. The contours appear polarized, the relative displacements between the stationary and active zones differ significantly, and a concentration of equal displacement lines appears between the active and stationary zones, where the sliding shear surface is generated [22]. The crack between neighboring movable beams develops, and a second crack forms (see Figure 8b), called the second arching surface. Meanwhile, the height of the soil arching structure increases to approximately 132 mm, significantly greater than the beam spacing, and the active zone remains rectangular. Finally, cracks in the arching and sliding surfaces connect to form a W-shaped shear band.

Figure 9 illustrates variations in the soil pressure on the beams. The soil pressure-displacement curves are similar to those in conventional trapdoor tests. When the relative displacement *δ* increases, the pressure on the movable beams increases significantly. The soil pressure on the stationary zone shifts to the movable beams, and the soil pressure on the fixed beams falls progressively. Several transitional phases and feature points are selected and specified (Figure 9) to characterize the development process of the passive soil arching effect. The feature points rely on the beam spacing, relative compactness, and water content [30]. The A, B, C, D, and E represent the beginning state, elastic limit, maximum pressure, point of abrupt change, and final state, respectively. The pressure evolution of passive soil arching was split into four typical phases based on selected feature locations: development stage (A~C), stabilization stage (C~D), local damage stage (D~E), and re-stabilization stage (E~F).

In the development stage, when *δ* ≤ 4 mm, growth in the relative displacement causes the backfill sand to undergo shear deformation, making the stress redistribution phenomena gradually noticeable, and the soil pressure on the movable beams increases approximately linearly. The pressure at SS3 increased faster than at SS4. When *δ* = 2 mm, the pressure on the movable beams approaches the elastic limit. The load achieves its maximum value when *δ* = 4 mm. In the stabilization stage, when 4 mm < *δ* < 8 mm, the increased pressure on the movable beams is minimal and generally steady, while the deformation of sand on the movable beams is plastic, and the local sand has yielded. The pressure at SS3 is greater than SS4.

In the local failure stage, when 8 mm ≤ *δ* < 10 mm, pressure growth on the movable beam changes abruptly, the pressure at SS3 begins to decrease, and the pressure at SS4 grows and is more prominent than that of SS3 while becoming stable. This could be because the soil arching generated by the upward movement of the movable beam can be separated into inner and outer arching (Figure 10). The arching closest to the I beam (Figure 10) is known as inner arching, while that closest to the O beam (Figure 10) is known as outer arching. During soil arching evolution, inner arching plays the primary role, and outer arching is secondary before the first sliding surface appears. With an increased relative displacement, the outer arching starts to play the leading arching role, and the bearing role of the inner arching weakens after the first sliding surface emerges. The extension of the arching structure implies a transfer of the load-bearing characteristics in the soil arching structure. In the re-stabilization stage, when 10 mm ≤ *δ* < 14 mm, after a period of local arching failure, a new arching structure is reconstructed, and the soil pressures are redistributed before eventually stabilizing.

### 3.2. Influence of Beam Spacing on Soil Arching Effect

Figure 11 illustrates soil arching contours at different *η* (*η* = *s*/*a*, ratio of beam spacing *s* to beam width *a*), showing the architecture and characteristics of the arching structure. Soil arching with *η* = 1 evolves differently than the other three groups. The arching structure is not evident at this beam spacing, and only the initial bending configuration forms between movable beams [16]. When *η* = 2 and 3, the sand above the movable beam creates nearly triangular arches with heights of 123 and 126 mm, respectively. The arching structure is relatively stable, and the entire active zone is trapezoidal. When *η* = 4, the arching structure is parabolic, but the entire structure is extremely unstable and prone to plastic damage near the foot. The final arch height is 130 mm, and the entire active zone is trapezoidal. Overall, the condition *η* = 1 cannot form a complete W-shaped shear band comparatively, and an increased beam spacing gradually expands the active zone area. However, the displacement of sand particles in the active zone gradually decreases. Moreover, the arch structure height grows with *η* and decreases with *ζ* (*ζ* = *H*/*s*) (Figure 12). Thus, beam spacing has little impact on the arch shape and shear zone, while the arch structure strength and height have a significant influence. The beam spacing greatly affects the arch shape, shear band, arch structure strength, and arch height.

Figure 13 shows pressure variations on the beams with *η*. When *η* ≥ 2, the pressure evolution on the beams adheres to the law described in Section 3.1. When *η* = 1, the pressure development on the movable beams differs. The pressure curves of the I and O beams are nearly the same and no longer change suddenly, indicating that the beams are loaded almost simultaneously, as shown in Figure 13a. The pressure results show that an increase in *η* causes the I beam to load before the O beam in the soil arching development stage, and the time interval gradually increases. Thus, the pressure change increases in the local failure stage. After re-stabilization, the pressure difference between the I and O beams also increases. For larger *η*, the average pressure on the movable beams increases, and the relative displacement required for the pressure to maximize increases (Figure 14). The *E* and *n* have little correlation with *η*. The *n* first decreases and then increases slowly, and *E* first increases and then decreases slowly (Figure 15). Therefore, the soil arching structure with a larger beam spacing can withstand greater external loads. The beam spacing significantly affects the soil pressure transfer and evaluation parameters (*E*, *n*).

Obviously, changes in the beam spacing greatly influence the evolution of the arching effect. The variation characteristics of the evaluation parameters indicate a stable arch structure is formed when *η* = 2~3.

### 3.3. Influence of Relative Density on Soil Arching Effect

Figure 16 shows soil arching contours with different *Dr* (δ = 12 mm), indicating aeolian sand with *Dr* ≥ 30% can form arching structures. When *Dr* = 30%, the aeolian sand particles are relatively loose, resulting in a low shear strength. The formed soil arching structure can be easily damaged at the foot, and the arch height is only 85 mm. When 50% ≤ *Dr* ≤ 70%, the relative density is moderate, the arching structure is more stable, the arching foot strength is greater, and the arch height and beam spacing are very close at 124 and 118 mm, respectively. When *Dr* = 90%, the relative density is larger, the particles are very dense, and only a tiny displacement is needed to form an arching structure. As the displacement increases, the bearing capacity of the arching structure becomes greater than the strength of the foot, causing damage. The final arching height is about 95 mm. In comparison, as the relative density increases, the area of the active zone expands, the average displacement increases, the arching height first increases and then decreases (Figure 17), the arching retains the same triangular shape, and the shear band is still W-shaped. Thus, the relative density has little impact on the arching shape and shear band but significantly affects the structural strength and arching height.

Figure 18 shows variations in the pressure on the beams with different *Dr*. When *Dr* ≤ 50% after the pressure on the movable beams reaches a maximum, it decreases faster and is almost without the stable stage. When *Dr* = 30%, the pressure reduces by nearly 80%, followed by *Dr* = 50%, which is reduced by 40%. The decreased rate of soil pressure on the I beam is greater than that on the O beam, which could be because the arching structure formed by the low relative density is unstable, and the stress transmission path is easily damaged, resulting in a sharp pressure drop at the movable beams. As the relative density increases when *Dr* = 70%, the pressure–displacement curve is consistent with the features in Section 3.1. When *Dr* = 90%, the pressure on the movable beams no longer produces mutations, but it only decreases slightly after the pressure reaches a maximum. The pressure on the I beam is greater than on the O beam, possibly because a greater relative density causes a larger overall shear strength of the arching structure. As the relative displacement increases, the inner arching continues to play a major bearing role until the foot is destroyed. The average pressure on the movable beams also increases with the relative density, but the relative displacement required for the maximum pressure decreases. When *Dr* = 30%, a relative displacement of about 4 mm is required for the pressure to reach its maximum. When *Dr* = 90%, only 2 mm is required (Figure 19), indicating that a high relative density can help form arching structures earlier. However, the stress concentration rate and load transfer efficiency for evaluating the degree of arching evolution remained nearly unchanged with the relative density (Figure 20). The relative density significantly affects the soil pressure transfer in the arching structure and formation time but has little effect on the evaluation parameters (*E*, *n*).

In summary, the arching structure formed by the medium and high relative density of aeolian sand is relatively stable and has a good stress concentration ratio and load transfer efficacy. (*Dr* = 30% is loose sand, *Dr* = 50% is medium dense sand, and *Dr* = 70% and 90% are dense sand.)

### 3.4. Influence of Water Content on Soil Arching Effect

Figure 21 shows the contours of soil arching at different water contents *ω* (*δ* = 12 mm). Aeolian sand with 3%≤ *ω* ≤9% can form arching structures. When *ω* = 3%, a stable arching structure is formed, and the footing strength is sufficient to support the load due to relative displacement and arching height of 121 mm. When *ω* ≥ 5%, the formed arching structure is prone to damage at the foot, and the arch height is small below 120 mm. An added test with *ω* = 0 shows that an arching structure cannot be formed. This may be because the adhesion of dry aeolian sand is nearly 0, and the grade is very poor, weakening the “wedge tightening” action of the particles. The friction between particles is also very small, resulting in the arching feet being insufficient to support the load coming down from above. In comparison, the extent of the active zone decreases with the water content, the average displacement of the sand decreases, and the arching height decreases (Figure 22). However, the shear bands are consistent and are all W-shaped. Thus, the water content has little impact on the arch shape and shear band but significantly affects the structure strength and height.

Figure 23 shows variations in the pressure on the beams at different water contents. The pressure–displacement curve is consistent with the feature in Section 3.1 when 3% ≤ *ω* ≤ 5% (low water content). When *ω* ≥ 7% (high water content), the pressure increases on the active beam is unstable, and the pressure difference between I and O beams gradually decreases. This may be because the apparent cohesion of the aeolian sand increases at greater water contents, while the internal friction angle decreases and the adsorption capacity increases. Thus, the arching structure becomes a single entity, and the inner and outer arches bear the loads nearly simultaneously. As the water content increases, the average pressure on the movable beams decreases, and the average pressure reduces by approximately 1 kPa for each 2% increase. However, the relative displacement required for the pressure to reach its maximum value increases. When *ω* = 3%, the relative displacement for the pressure to reach the maximum is about 4 mm; when *ω* = 9%, the required relative displacement is greater (see Figure 24). This explains why the formation of arching structures in aeolian sand at high water content occurs relatively late. In addition, as the water content increases, the stress concentration ratio increases, and the load transfer efficiency decreases, which are stable at 0.28 and 0.81, respectively. Therefore, high water contents can weaken the pressure transmission in the aeolian sand arching structure (Figure 25). The water content significantly affects the soil pressure transfer in the arched structure, the formation time of the arched structure, and the evaluation parameters (*E*, *n*).

According to the above description, when 3% ≤ ω ≤ 5% (the low water content), the load transfer is sufficient and the soil arch evolves well.

## 4. Conclusions

A series of trapdoor tests were performed to explore the passive soil arching effect in aeolian sand backfills for grillage foundations. The geometric characteristics of the arching structure and changes in the displacement field were observed via PIV, and the miniature dynamic soil stress sensors monitored pressure changes. This study focused on the evolution pattern of the soil arching morphology and the development of soil pressure. The influence of the trapdoor width, the sand relative density, and water content on the passive soil arching effect is also discussed. The main conclusions are as follows:

(1) During the arching structure evolution, an increased relative displacement gradually extends the influence of the aeolian sand deformation to the backfill surface and stationary zone. Two arching surfaces are successively generated between the movable beams to form a triangular arch structure with a height that increases significantly. Sliding surfaces are generated at the boundary between the active and stationary zones. Finally, cracks in the arching surface and sliding surface connect to form a W-shaped shear band.

(2) According to the soil pressure development characteristics, the evolution of the passive soil arching effect can be split into four stages: development, stabilization, local damage, and re-stabilization. The soil pressure increases rapidly at first and then tends to stabilize. Still, there is a significant difference in the bearing capacity of the inner and outer arches, with the inner arch playing a major role. As the second arching surface appears, a new arching structure is reconstructed, the soil pressure is redistributed, and the outer arch plays a major role.

(3) The *η* significantly affects the arch shape, shear band, soil pressure transfer, and evaluation parameters (*E*, *n*). A smaller *η* forms the initial bending configuration with an unobtrusive arching structure, and the inner and outer arches are loaded almost simultaneously. As *η* increases, the arch shape changes from a triangle to a parabola, and the inner arch is loaded earlier. The sudden change in soil pressure is greater, the arch height increases, *n* first decreases and then increases, and *E* first increases and then decreases. When *η* = 1, there is no complete shear band, unlike the others.

(4) The *Dr* has little impact on the arch shape, shear band, and evaluation parameters (*E*, *n*), but it significantly affects the arching structure strength, arch height, and soil pressure transfer. As *Dr* increases, the arch height initially increases and then decreases. The arching structure produced by low *Dr* is easily damaged at the foot, wherein the soil pressure increases and then decreases rapidly. As *Dr* increases, the arching structure is stronger when produced by medium and high *Dr*. Then, sudden changes in the soil pressure are reduced, and the arch height increases and then decreases.

(5) The *ω* has little impact on the arch shape and shear band but significantly affects the arch height, evaluation parameter (*E*, *n*), and soil pressure transfer. As *ω* increases, the arch height and *E* decrease while *n* increases. The arching structure produced by the low water content has a greater strength. The soil pressure changes abruptly or remains stable at the appearance of the first arching surface, and the structure forms earlier. There is adhesion in the arching structure caused by the high water content, and the soil pressure does not decrease after reaching its maximum but instead weakens the soil pressure transmission.

## Figures and Tables

**Figure 1 sensors-23-08098-f001:**
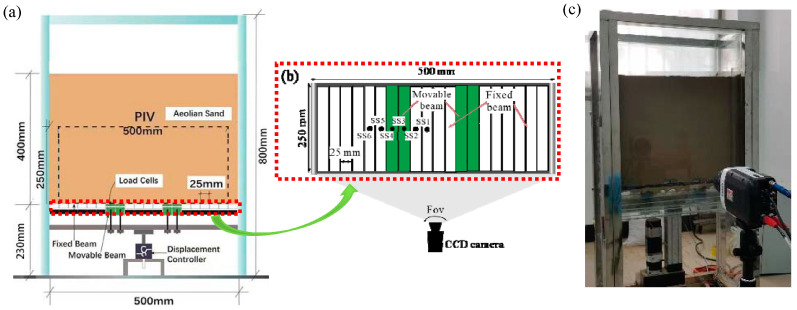
Scheme of the trapdoor apparatus: (**a**) cross-section view, (**b**) plan view, and (**c**) photo.

**Figure 2 sensors-23-08098-f002:**
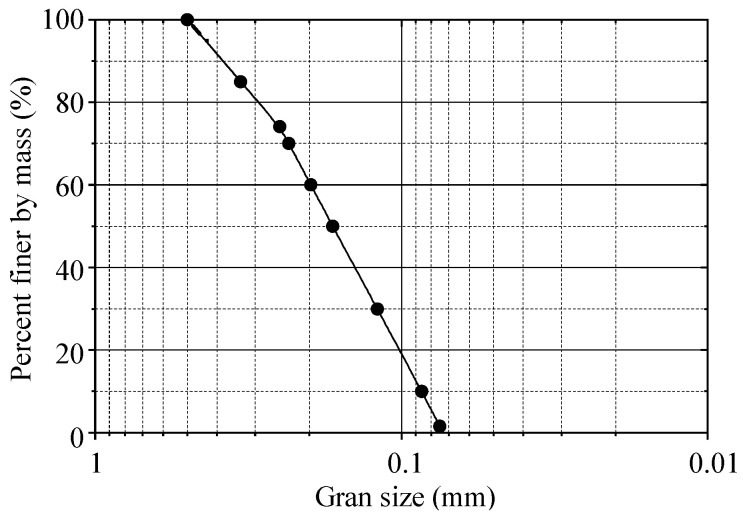
Grain size distribution of aeolian sand.

**Figure 3 sensors-23-08098-f003:**
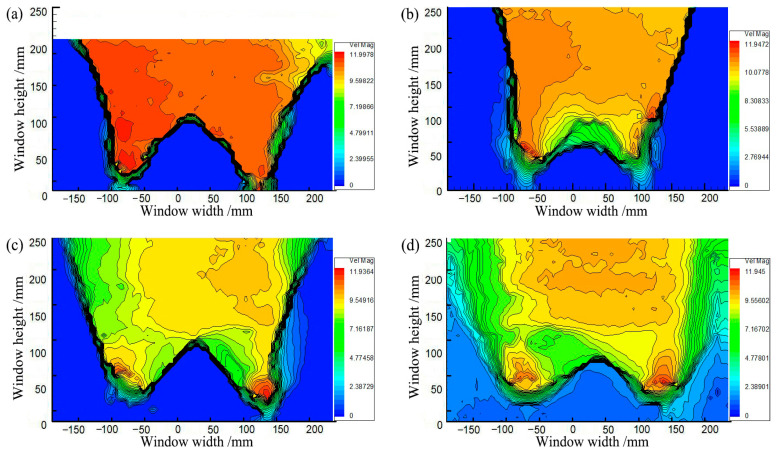
Contours of soil arching: (**a**) *H* = 200 mm, (**b**) *H* = 300 mm, (**c**) *H* = 400 mm, (**d**) *H* = 500 mm.

**Figure 4 sensors-23-08098-f004:**
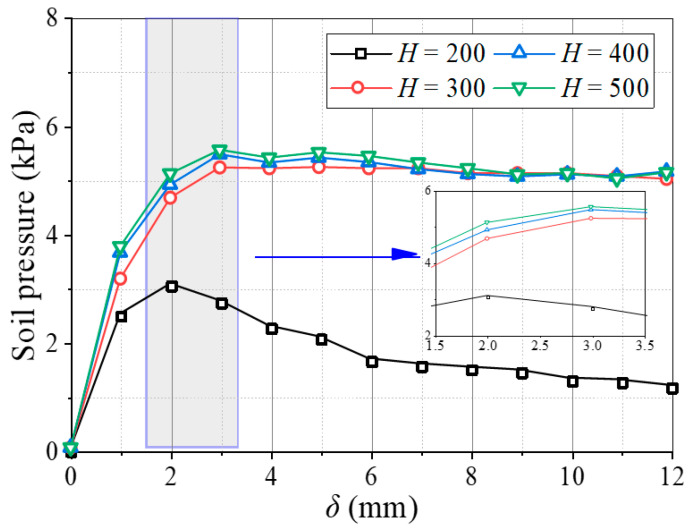
Variation in the average soil pressure on movable beams at different backfill heights.

**Figure 5 sensors-23-08098-f005:**
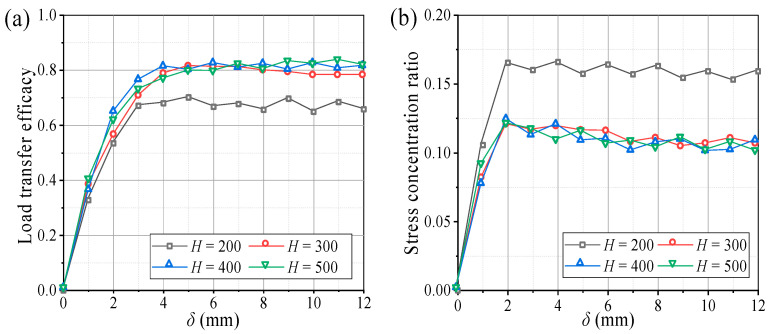
Characteristic curves of evaluation parameters at different backfill heights. (**a**) Load transfer efficacy (*E*) and (**b**) stress concentration ratio (*n*).

**Figure 6 sensors-23-08098-f006:**
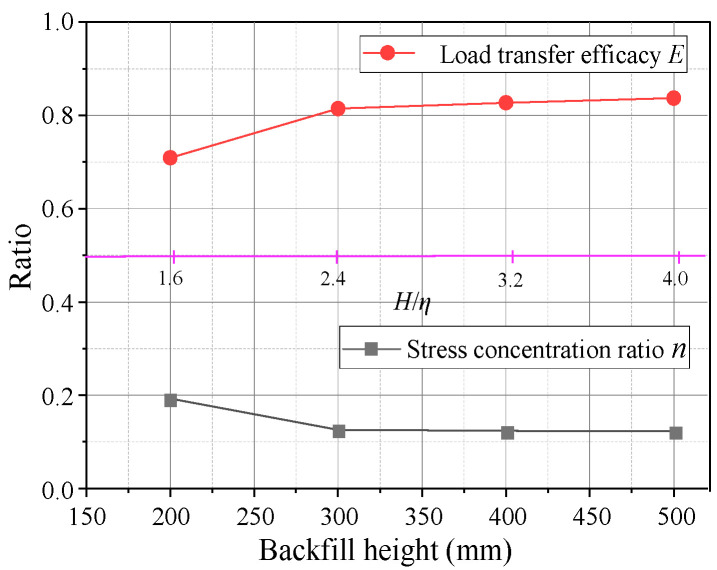
Variations of *E* and *n* with the backfill height.

**Figure 7 sensors-23-08098-f007:**
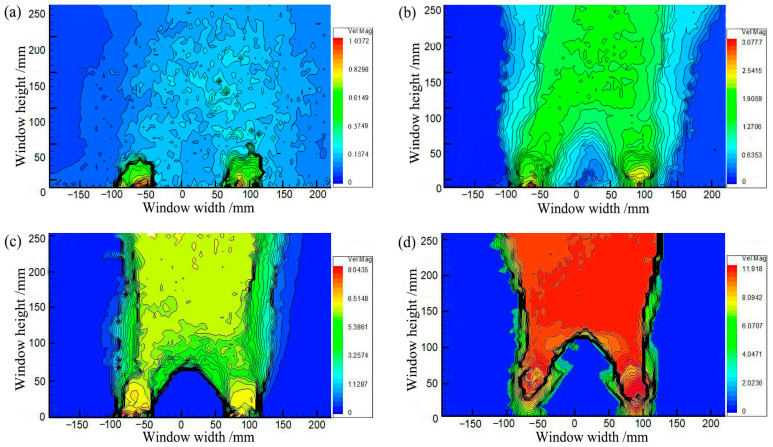
Contours of soil arching for (**a**) *δ* = 2 mm, (**b**) *δ* = 4 mm, (**c**) *δ* = 8 mm, and (**d**) *δ* = 12 mm.

**Figure 8 sensors-23-08098-f008:**
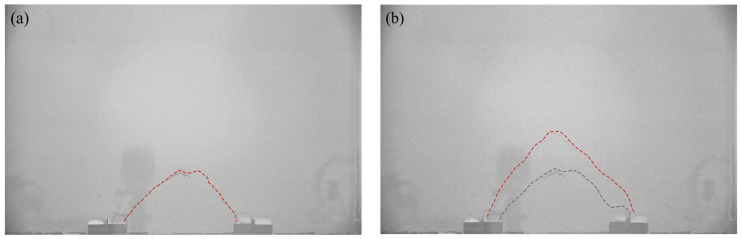
Arching surface evolution diagrams. (**a**) First arching surface and (**b**) second arching surfaces.

**Figure 9 sensors-23-08098-f009:**
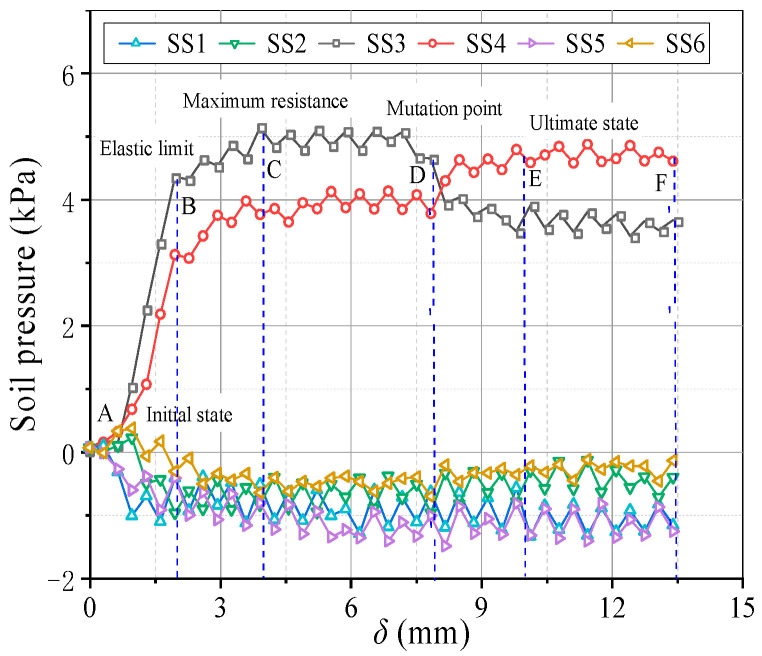
Variations in the soil pressure on the beams at the various sensor locations.

**Figure 10 sensors-23-08098-f010:**
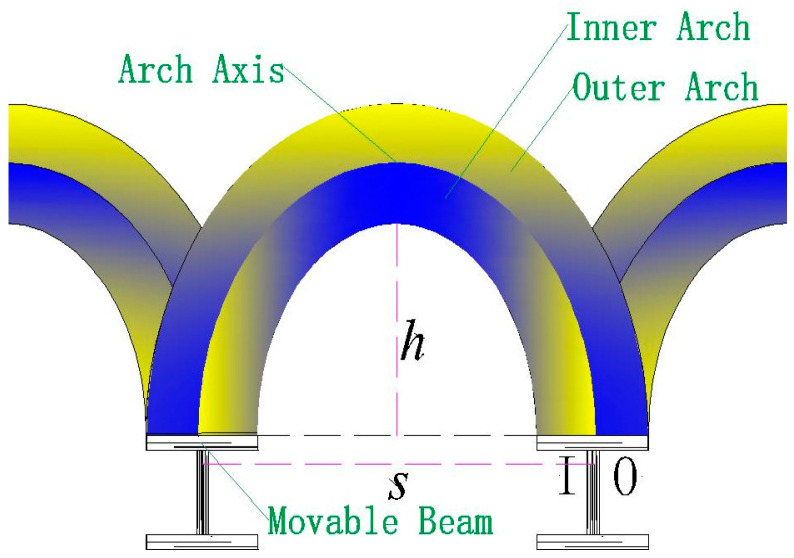
Diagram of the soil arching structure between movable beams.

**Figure 11 sensors-23-08098-f011:**
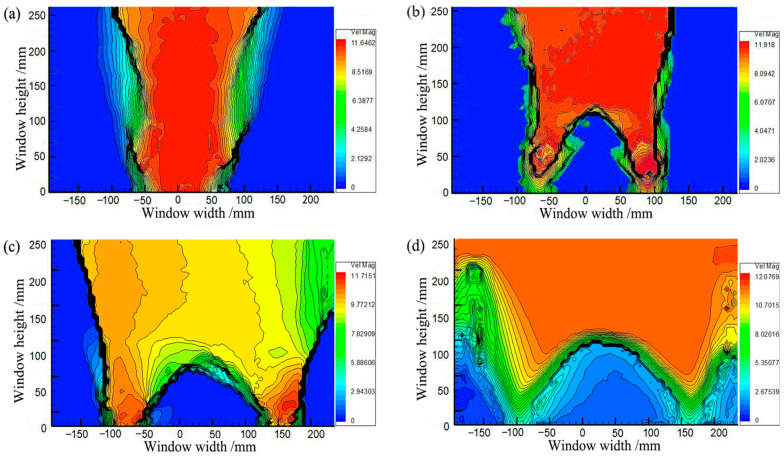
Contours of soil arching. (**a**) *η* = 1, (**b**) *η* = 2, (**c**) *η* = 3, and (**d**) *η* = 4.

**Figure 12 sensors-23-08098-f012:**
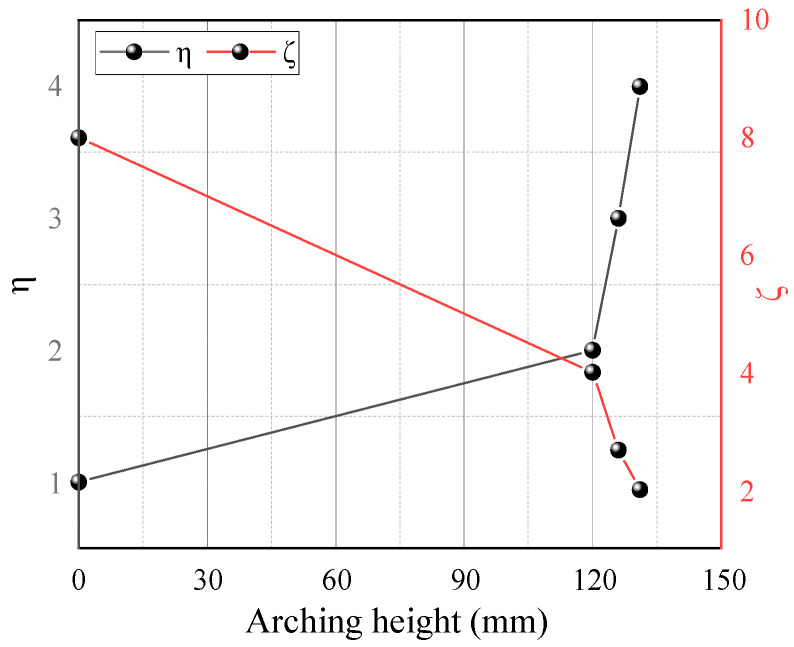
Variations in the arching height with *η* and *ζ*.

**Figure 13 sensors-23-08098-f013:**
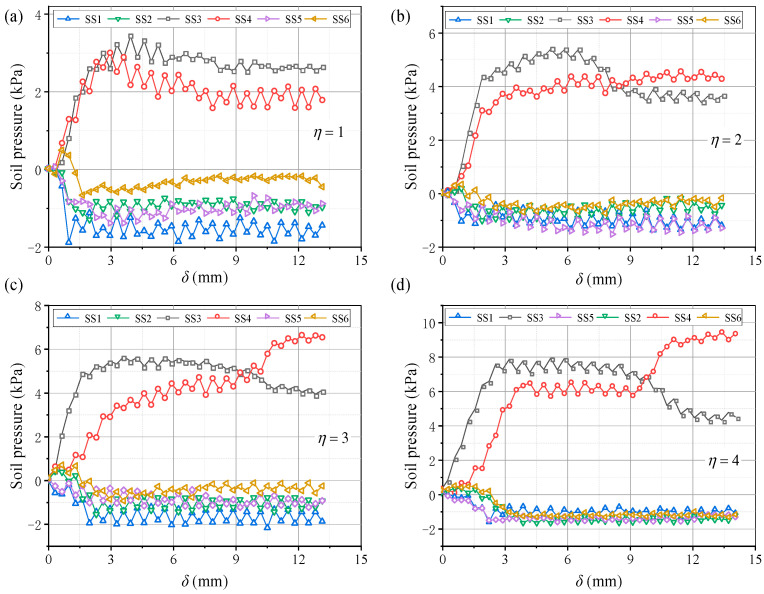
Variations in the soil pressure for (**a**) *η* = 1, (**b**) *η* = 2, (**c**) *η* = 3, and (**d**) *η* = 4.

**Figure 14 sensors-23-08098-f014:**
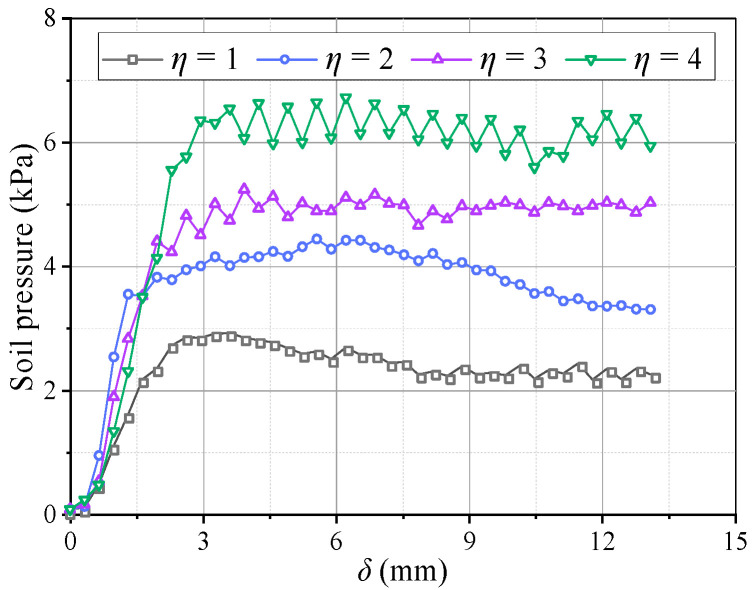
Variations in the average soil pressure on movable beams at different *η*.

**Figure 15 sensors-23-08098-f015:**
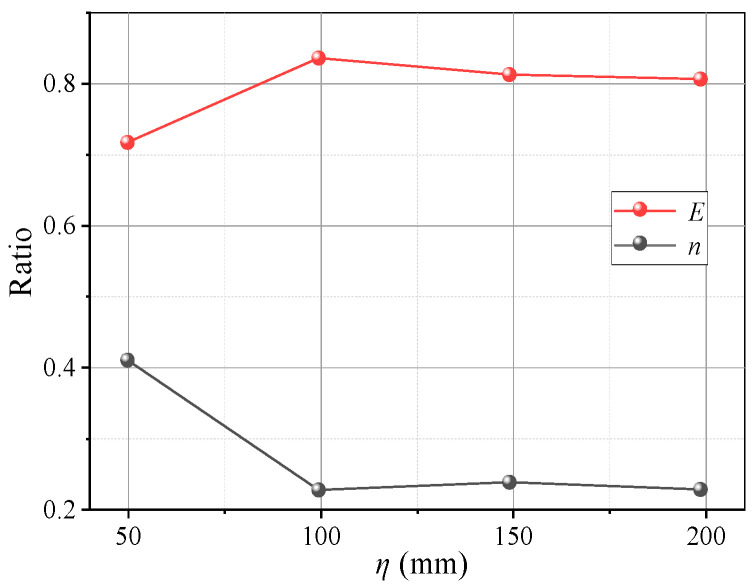
Variations in *E* and *n* with *η*.

**Figure 16 sensors-23-08098-f016:**
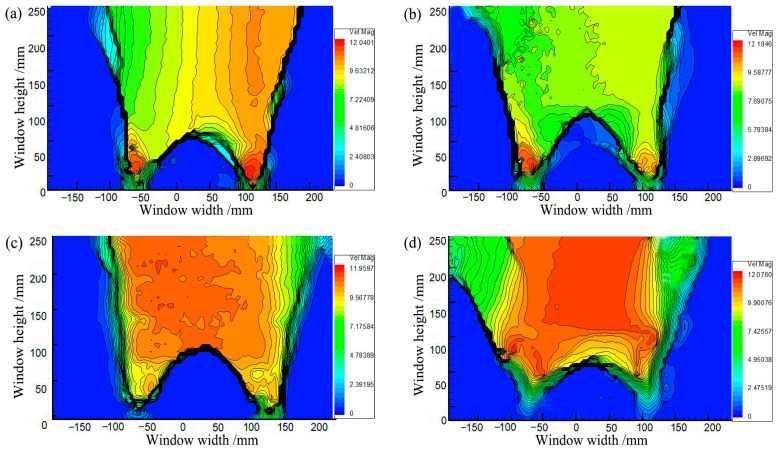
Contours of soil arching for (**a**) *Dr* = 30%, (**b**) *Dr* = 50%, (**c**) *Dr* = 70%, and (**d**) *Dr* = 90%.

**Figure 17 sensors-23-08098-f017:**
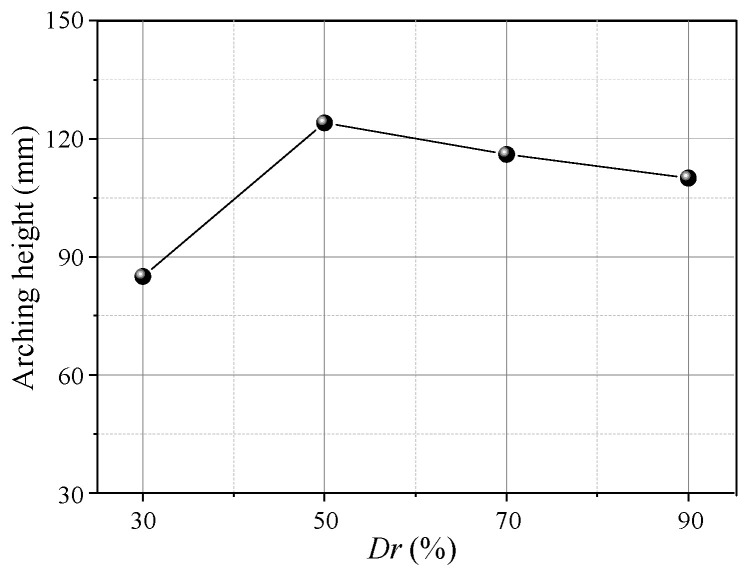
Variation in the arching height with *Dr*.

**Figure 18 sensors-23-08098-f018:**
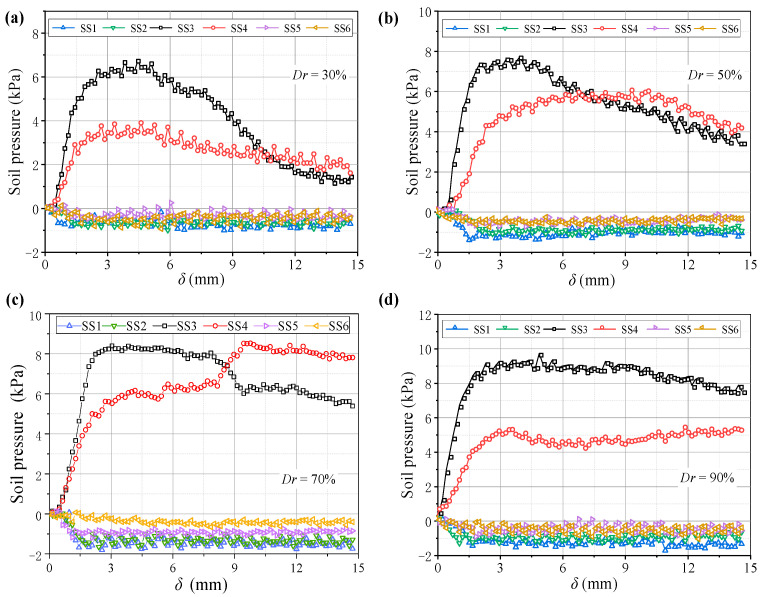
Variations in the soil pressure with (**a**) *Dr* = 30%, (**b**) *Dr* = 50%, (**c**) *Dr* = 70%, and (**d**) *Dr* = 90%.

**Figure 19 sensors-23-08098-f019:**
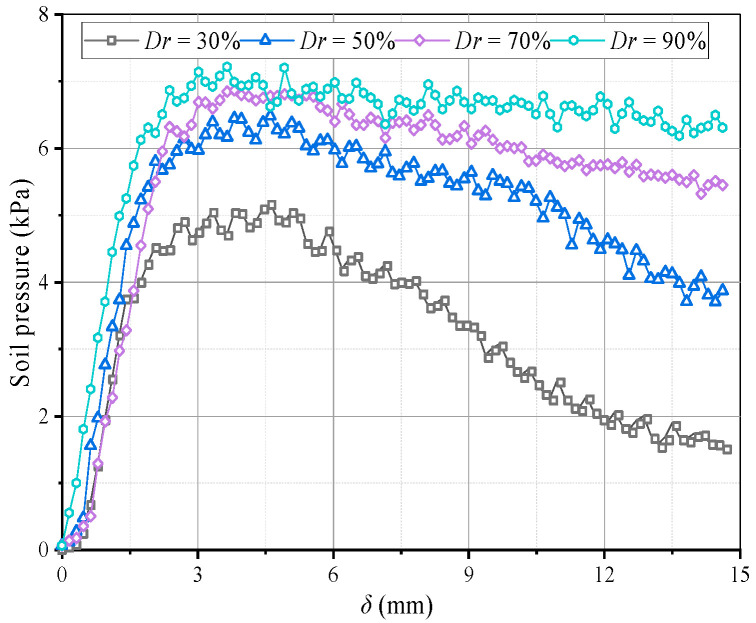
Variations in the average soil pressure on movable beams at different *Dr*.

**Figure 20 sensors-23-08098-f020:**
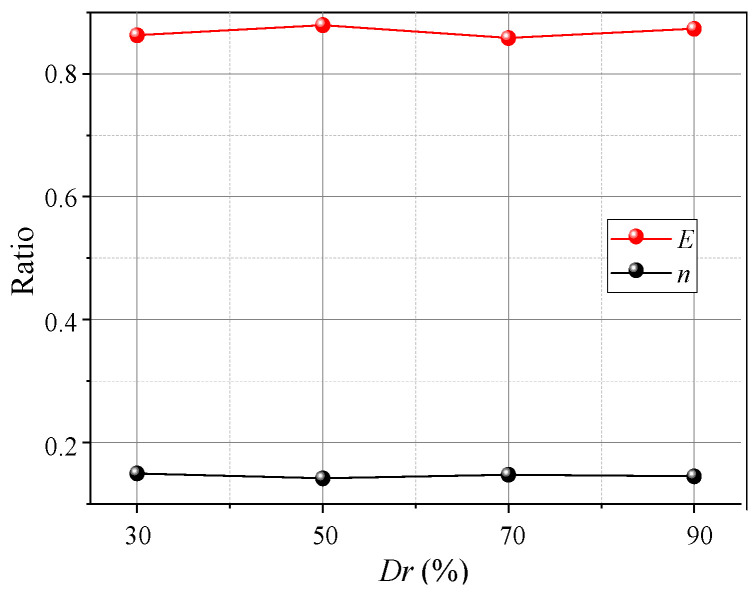
Variations in *E* and *n* with *Dr*.

**Figure 21 sensors-23-08098-f021:**
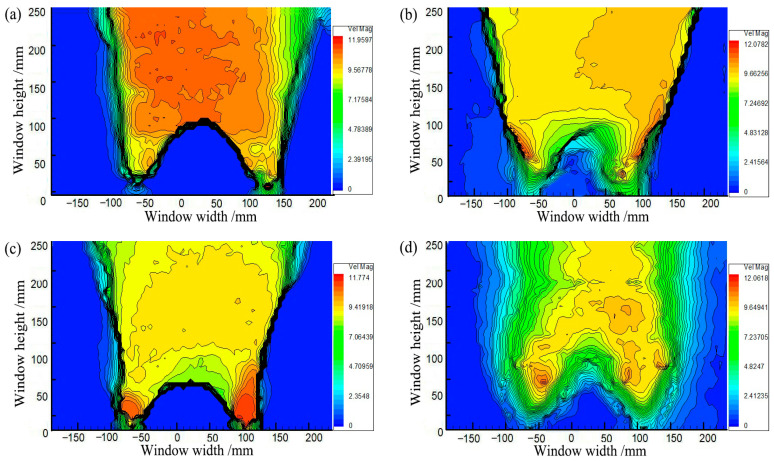
Contours in the soil arching for (**a**) *ω* = 3%, (**b**) *ω* = 5%, (**c**) *ω* = 7%, and (**d**) *ω* = 9%.

**Figure 22 sensors-23-08098-f022:**
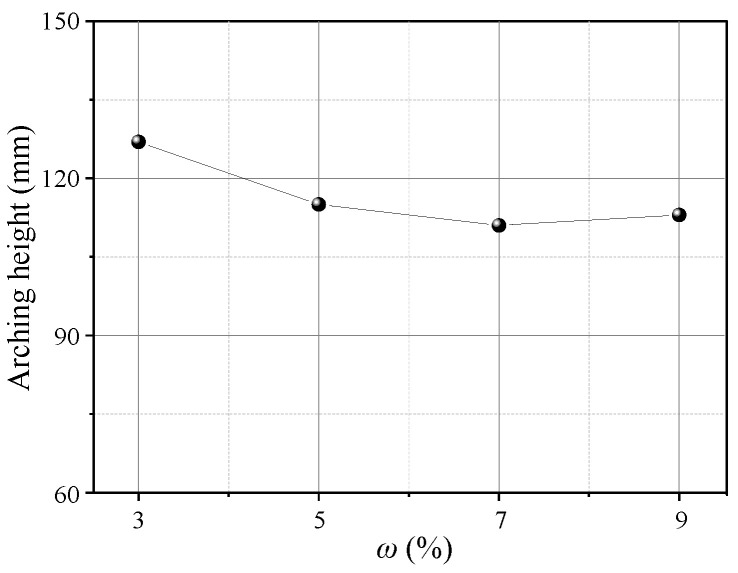
Variations in the arching height with *ω*.

**Figure 23 sensors-23-08098-f023:**
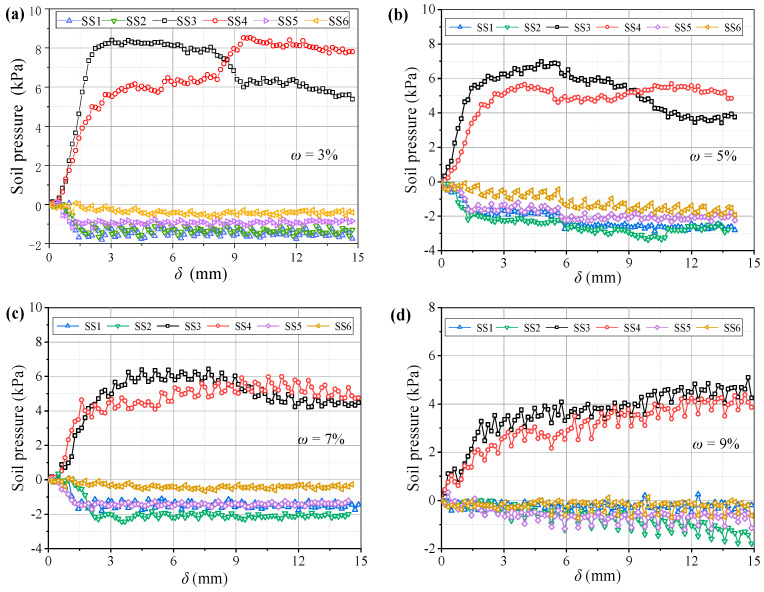
Variations in the soil pressure with (**a**) *ω* = 3%, (**b**) *ω* = 5%, (**c**) *ω* = 7%, and (**d**) *ω* = 9%.

**Figure 24 sensors-23-08098-f024:**
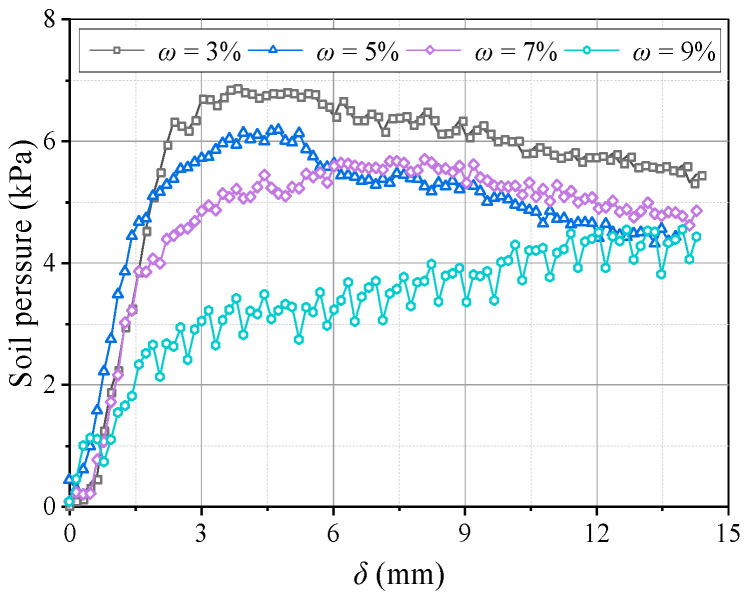
Variations in the average soil pressure on movable beams at different *ω*.

**Figure 25 sensors-23-08098-f025:**
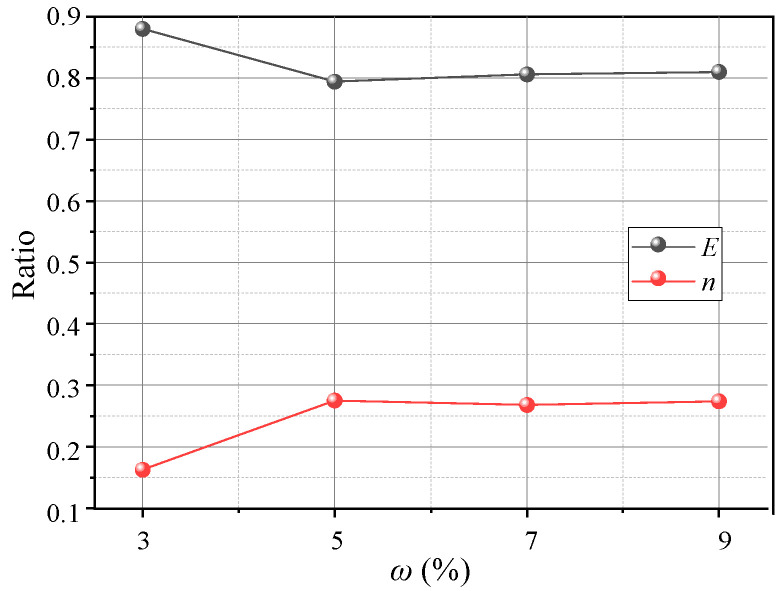
Variations in *E* and *n* with *ω*.

**Table 1 sensors-23-08098-t001:** Characteristics of the collected aeolian sand.

Characteristics	Value
Water content *ω* [%]	2.8
Void ratio *e* [-]	0.73
Specific gravity *Gs* [-]	2.67
Density *ρ* [kg/m^3^]	1.74
Relative density *Dr* [%]	68.8
Angle of shearing resistance *φ* [°]	40.8

**Table 2 sensors-23-08098-t002:** Mineral composition of the collected aeolian sand.

The Proportion of Minerals				
Quartz	Plagioclase	Potash Feldspar	Dolomite	Illite
55.2%	28.4%	15.6%	0.6	0.2

**Table 3 sensors-23-08098-t003:** Chemical composition of the collected aeolian sand.

SiO_2_	Al_2_O_3_	K_2_O	CaO	Na_2_O
72.33%	14.37%	3.50%	3.02%	2.67%
Fe_2_O_3_	MgO	TiO_2_	P_2_O_5_	BaO
1.85%	1.32%	0.49%	0.13%	0.100%
SO_3_	SrO	MnO	ZrO_2_	Rb_2_O
0.063%	0.057%	0.040%	0.019%	0.012%

**Table 4 sensors-23-08098-t004:** Test conditions.

Test Cases			Backfill Height *H* (mm)	Trapdoor Width	Ratio of Filling Height to Trapdoor Width *H*/*s* (-)	Relative Density *D_r_* (%)	Water Content *ω* (%)
Group	No.	Code		*s* (mm)			
I	1	F1	200	125	1.60	70	3
	2	F2	300	125	2.40	70	3
	3	F3	400	125	3.20	70	3
	4	F4	500	125	4.00	70	3
II	5	T1	400	50	8.00	70	3
	6	T2	400	100	4.00	70	3
	7	T3	400	150	2.67	70	3
	8	T4	400	200	2.00	70	3
III	9	M1(R3)	400	125	3.20	30	3
	10	M2	400	125	3.20	50	3
	11	M3	400	125	3.20	70	3
	12	M4	400	125	3.20	90	3
IV	13	R1	400	125	3.20	70	5
	14	R2	400	125	3.20	70	7
	15	R4	400	125	3.20	70	9

## Data Availability

Not applicable.

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
