# Peer review of "Passive Soil Arching Effect in Aeolian Sand Backfills for Grillage Foundation"

_sensors, 2023, doi:10.3390/s23198098_

Round 1

Reviewer 1 Report

The submitted manuscript is well presented in general however, formatting and presentation of the manuscript is recommended.

minor editing required around incorrect use of capitalization of certain words however, the overall quality of English Language is sound. 

Reviewer 2 Report

The article mainly studies the evolution mechanism and influencing factors of the passive soil arching effect is crucial to understand its working principle, It has a certain degree of innovation, but there are still the following issues that need to be modified before the article is accepted for publication, recommended as a reviewer for publication after major repairs:

1. The abstract and introduction of the paper need to further emphasize the research significance of this article, so that readers can better find the focus of the article's research;

2. The author's research conclusion needs to be further simplified and reduced, highlighting the more meaningful conclusions of the paper;

3. The article needs to add more references related to the content of the paper.

4. The English language issue of the article needs further improvement to promote the improvement of the English proficiency of the article;

5. There are also a large number of language spelling errors and Syntax error in the paper. It is suggested that the author carefully check and correct them.

The article mainly studies the evolution mechanism and influencing factors of the passive soil arching effect is crucial to understand its working principle, It has a certain degree of innovation, but there are still the following issues that need to be modified before the article is accepted for publication, recommended as a reviewer for publication after major repairs:

1. The abstract and introduction of the paper need to further emphasize the research significance of this article, so that readers can better find the focus of the article's research;

2. The author's research conclusion needs to be further simplified and reduced, highlighting the more meaningful conclusions of the paper;

3. The article needs to add more references related to the content of the paper.

4. The English language issue of the article needs further improvement to promote the improvement of the English proficiency of the article;

5. There are also a large number of language spelling errors and Syntax error in the paper. It is suggested that the author carefully check and correct them.

Reviewer 3 Report

The work reported in the paper presents the evolution of passive soil arching effect of aeolian sand through model-scale laboratory tests. The novelty is that the trapdoor test is conducted through upward motion into the sand. After reading the manuscript in its entirety, this reviewer feels that the paper in the current form should be revised to clarify the following points:

1. Passive soil arching can be widely found in practice. For example, rigid buried pipe experiences passive soil arching, and the soil prism loading is increased. Flexible buried pipe experiences active soil arching, and the soil prism loading is decreased. Active soil arching is beneficial to the design of geostructures, and normally researchers try to figure out a measure to convert passive soil arching into active soil arching by introducing compressible materials (Geosynthetics International, 2017, 24(6): 615-624; 2018, 25(5): 494-506). The scenario that the authors simulated should be clarified, or please explain when active soil arching can be observed and how it can be analyzed.

2. The model container is 500 mm in length, and the length of the trapdoor is 250 mm. The definitions of length for the model container and the trapdoor are different, i.e., the two are perpendicular. One should define the length scale using the same coordinates. How do you determine the spacing between two moving beams or the width of the trapdoor? All these dimensions can affect the scale effect or cause the boundary effect. From the subsequent Fig. 3, it seems that the results are heavily affected by the boundary effect. This suggests that either the model container is too small or the setting of the trapdoor is inappropriate.

3. More details on PIV should be provided. For example, how do you set the focal length and the distance between camera and window? How do you minimize the measurement error induced by lens distortion? What is the estimated accuracy in displacement (in mm and/or size of each pixel) that the system can calculate? As long as all these are reported, it can establish the confidence for the readers regarding the accuracy of the analysis (Géotechnique, 2003, 53(7): 619-631; 2016, 66(4): 275-286; 2018, 68(1): 1-17; Canadian Geotechnical Journal, 2015, 52(9): 1199-1220; 2016, 53(5): 727-739).

4. Table 1: Please explain the source of cohesion in sand. Please also clarify whether the friction angle of 28.5 degrees is the peak or the constant volume friction angle. It seems that very loose materials can have such low friction angle. However, the current study used the relative density of 68.8%, which is medium dense sand. For such density, one can expect that the friction angle can be around 35 degrees.

5. Some typos can be seen, e.g., eolian sand. “Cloud atlas” should be revised as “contours”. “Fissure” is a word that is frequently used by rock mechanics engineers, rather than geotechnical engineers. Corrections should be made thoroughly.

6. Fig. 3: The authors comment on the presence of uneven settlement as a function of soil depth. However, the term of equal settlement plane is never mentioned (Canadian Geotechnical Journal, 2023, 60(6): 802-816).

7. The derived soil arching ratios should be compared with existing prediction models, instead of presenting the trench of the measurements only (International Journal of Geomechanics, 2017, 17(12): 04017112; 2018,18(6): 04018056; Soils and Foundations, 2019, 59(6): 2206-2219).

8. The capacity and the accuracy of SST need to be provided.

9. Fig. 8: The presence of arching surface should be highlighted or marked in the figure.

10. “SS3 soil pressure was more excellent than SS4 soil pressure.” The statement is not clear.

11. Regarding the influence of water content, it is advised to make calculations for matric suction, and comment on the changes in shear strength induced by matric suction. In this way, the pattern of failure mechanism affected by matric suction can be identified, rather than using the term of adhesion (can hardly be determined quantitatively).

See comments

Reviewer 4 Report

The evolution and Influencing factors of the passive soil arching effect through a trapdoor device and Particle Image Velocimetry (PIV) technology were investigated. The study is well structured. The comments are listed as follows.

(1) The passive soil arching structure has been developed by other researchers according to literatures. What are the differences and the shining point in the study?

(2) The behaviour of sands under loading effect was conducted in this paper. The displacement is closely related with the damage characteristics, which could be analysed with the analytical model (Plz ref to https://doi.org/10.1142/S1758825123500369). The results of shear damage analysis can provide a fundamental basis for stability analysis in geotechnical engineering. The shear damage model and characteristics can be reviewed and added in the introduction or relative section.

(3)  The photos of the trapdoor apparatus can be added.

(4) What is the accuracy of CCD camera and observed displacement?

(5) What is the meaning of H in figure title? The first appearance of variables should be illustrated, such as H and s.

(6) More recent studies of other monitoring techniques, such as the borehole observation and displacement monitoring, in the explanation of the damages in geotechnical structures can be reviewed.

(7) The reason for the fluctuation of data points in Figure 13, etc. can be discussed.

Reviewer 5 Report

1.         The references cited in the introduction are related to the passive soil arch effect, but there are few suggestions to supplement the relevant literatures on its evolution and influencing factors.

2.         The scale of Figure a in Figure 1 should be re-labeled.

3.         The background and reasons for choosing soil samples from Yulin area as backfill materials are not described.

4.         Table 3 examines the spelling of all chemical ingredient names, paying particular attention to the presence of redundant punctuation.

5.         Check whether the order of tables in the full text is repeated. For example, two tables 3 are suggested to be modified.

6.         In the part of results and analysis, the author only focuses on the description of the experimental results and the data shown in the figure, and the analysis of the results is not enough to suggest supplements.

7.         The overall structure of the article is slightly confused.

8.         The conclusion part at the end of the paper is not full enough to summarize the results, and it is suggested to rewrite it.

9.         There are obvious grammatical errors in the text. It is suggested that the author re-read the whole text and correct the grammatical errors in the text.

10.      The references in this paper are generally old, so it is suggested that the author cite more recent references.

Moderate editing of English language required

Round 2

Reviewer 2 Report

The article has been carefully revised and is ready for publication

The article has been carefully revised and is ready for publication

Reviewer 3 Report

Good revisions are done, and the revised version can be accepted. 

Reviewer 4 Report

Thanks for the revision.